# Selected Partially Labeled Learning for Abdominal Organ and Pan-cancer Segmentation

Yuntao Zhu[1(✉)][0000−0003−2816−2709], Liwen Zou[1][0000−0003−4085−4003], Linyao Li[1][0009−0001−2630−1683], and Pengxu Wen[1][0009−0000−5211−4876]

[1] Department of Mathematics, Nanjing University, Nanjing, China
YuntaoZhu7@smail.nju.edu.cn

**Abstract.** Obtaining labeled data from medical images is very expensive and labor intensive. At the same time, the large number of existing publicly available medical image datasets are usually labeled with only some of the organs as target regions, while other organs in the image are ignored. It is a challenge to train a neural network to segment all labeled categories using only partially labeled data. We design a compound loss, the selected partially cross entropy and dice loss, that allows the neural network to learn specific categories from partially labeled data. In addition, we improve the inference and training process of nnU-Net to reduce computational resources and accelerate inference. Experiments demonstrate that our method achieves the average Dice Similarity Coefficient of 0.8514 and 0.1514 on 13 abdominal organ and tumor segmentation tasks, and enables the network to efficiently segment specific categories from partially labeled data. Moreover, it significantly improves the inference speed, with an average running time of 21.8 seconds, and uses only an average of 2531 MB of maximum GPU memory.

**Keywords:** Partially labeled learning · Accelerate inference · Lightweight network.

## 1 Introduction

Medical image segmentation aims to extract and quantify regions of interest in biological tissue or organ images. The results of target organ segmentation have many important clinical applications, such as organ quantification, surgical planning, and disease diagnosis. In recent years, deep learning-based methods have been widely used to automatically segment abdominal organs. Among these methods, nnU-Net [11] is a popular and robust framework that has won a number of organ segmentation challenges. Although it is convenient for fully supervised organ segmentation tasks and provides a solid baseline result by automatically setting network hyperparameters, this approach does not support weakly supervised segmentation and the inference process is computationally expensive and time consuming. Numerous studies have shown that the methodological performance of deep neural networks often relies heavily on the availability of large,

high-quality labeled datasets for organ segmentation tasks. In order to learn robust data representations for robust and efficient medical image segmentation, we need large datasets with thousands of labeled or unlabeled data for supervised, weakly supervised, and self-supervised learning. But, the annotation of 3D medical images is a difficult and laborious task. Thus, depending on the task, only a bare minimum of images and target structures is usually annotated. This results in a situation where a zoo of partially labeled datasets is available to the community. In this context, the organizer of FLARE2023 build a large-scale and diverse abdomen CT dataset, including 40000 CT scans from servel medical datasets. There are 2200 labeled data and 1800 unlabeled data available. Compared with FLARE 2021-2022 [17,18], the challenge for FLARE 2023 is how to leverage the large amount of partial labels and unlabeled data to improve the segmentation performance while taking into account efficient inference.

In recent years, there has been a rapid evolution of semi-supervised and self-supervised learning methods [31,24]. These techniques typically learn better representations by utilizing unlabeled data, ultimately improving segmentation performance. On the one hand, one frequently employed approach in semi-supervised learning is pseudo-labeling. This method pairs the segmentation results of the network on unlabeled data as pseudo-labels, adds them to the training set, and repeats the process over several iterations. On the other hand, integrating potentially valuable additional information from different datasets, which are partially labeled, can provide more information about different anatomical target structures or related details, as well as different types of pathology. Therefore, recent advances in weak supervision explore how partially annotated datasets can train a model to segment all annotated categories [12]. Early methods considered unlabeled organs as background [21,4] and imposed penalties for overlapping predictions based on mutual exclusivity of organs [22,5]. [26] transforms the cross-entropy loss and dice loss by assigning unlabeled data from partially labeled data to the background class. [3,30,13] predict just one structure of concern per forward pass through the integration of category information at various network stages. [14] use of partial cross entropy and intraclass gray regular terms allows segmentation under weak supervision. [25] ignores the channels where unlabeled categories are located, designs a loss function that mixes binary cross-entropy and dice loss, and can handle the task of category overlap in partial labeling learning. However, there is a lack of methods that utilize both pseudo-labeled data and partially labeled learning techniques to handle organ, tumor segmentation tasks like FLARE23 that contain partially labeled and unlabeled data.

In this paper, we present a framework that utilizes both pseudo-labeled and partially labeled learning by designing a selected partially loss. We also improve nnU-Net for efficient inference and less computational resource respectively. Specifically, we choose to merge 13 organ classes of pseudo-labels and partial labels, while leaving the remaining classes unchanged, resulting in a partial labeling of the tumor. The selected partially loss, which is a combination of cross-entropy loss and dice loss, introduces a selected class mask to deter-

mine whether the class loss will compute and backward gradient. Otherwise, we find that the resampling process in the inference is time-consuming. To address this issue, we have rewritten the implementation of the resampling method and utilized a smaller network and lower resolution to minimize the computational requirements during inference.

Our main contributions are summarized as follows:

– We present a new approach, selected partially loss, which enables the use of both pseudo label and partial label data, thereby expanding the potential applications of current segmentation models.
– We optimize the time-consuming components of the resampling code in nnU-Net.
– The experiment shows that our method improves the detectability of the segmentation network for the selected class. This outperforms the baseline by 5 percentage points for the Dice Similarity Coefficient (DSC).

## 2    Method

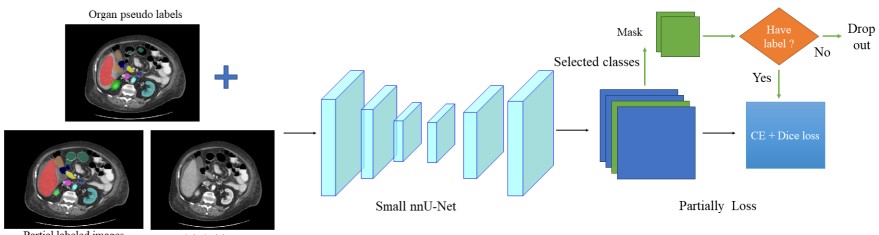

**Fig. 1.** Overview of our framework. Our framework consists of three parts. Firstly, we construct a training set by combining pseudo-labels. Secondly, we reduce computation costs by using a small nnU-Net. Lastly, we train nnU-Net by a selected-partially-loss so that it can learn from both unlabeled and partially-labeled data.

### 2.1    Preprocessing

For image prepossessing, all of our settings follow the default nnU-NetV2.

– Statistical analysis is conducted on data pertaining to volume spacing and foreground intensity.
– CT images are clipped at the 0.5 and 99.5 percentiles of foreground voxels.
– All images are normalized through the subtraction of the mean and division by the standard deviation.
– The volume is then resampled to a target spacing of (2.42,1.95,1.95).

## 2.2    Proposed Method

In Figure 1 we present an overview of our framework, which consists of three components. We filter the data by pseudo and select 300 cases as the training set. We then train a small nnU-Net using a compound partially loss on lower resolution. And our compound partially loss main refer to [25,26,14].

**Fusion of pseudo-labels and partial-labels** We use two pseudo-labels generated by [10,27], consisting of 13 organ categories for all 4000 cases. First, We calculate the DSC of the two pseudo-labels, evaluate their differences, and filter out the samples with DSC greater than 0.85. we sort them by their ID numbers. Subsequently, we select first 200 cases from partially labeled CT volumes and first 100 cases from unlabeled CT volumes to construct the training set. Then, the pseudo-labels are merged with the selected cases that do not contain the ground truth annotation of the class. Therefore, for the 300 cases, there are 13 organ labels (ground truth or pseudo) and tumor is partially labeled. All of our results use the pseudo-labels generated by the two FLARE 2022 methods.

**Problem definition** We begin with a dataset $D$, with $N$ image and label pairs $D = \{(x,y)_1, ..., (x,y)_N\}$. In the dataset, every image voxel $x_i, i \in [1, I]$, is assigned to one class $c \in C$, where $C$ is the label set associated to dataset $D$. Since the tumor is included in some organs commonly, but the pseudo label does not annotate the tumor. This implies that the network must predict multiple classes for one voxel to account for the inconsistent class definitions. To resolve the issue of label inconsistency, we separate the segmentation results for each class by applying a sigmoid activation function to replace the softmax activation function on the dataset.

**Partially loss for selected categories** We employ the binary cross-entropy (BCE) loss and the dice loss for each class over all $B, b \in [1, B]$, images in a batch:

$$L_c = \frac{1}{B \times I} \sum_{b,i} BCE(\hat{y}_{i,b,c},\ y_{i,b,c}) - \frac{2 \sum_{b,i} \hat{y}_{i,b,c}\, y_{i,b,c}}{\sum_{b,i} \hat{y}_{i,b,c} + \sum_{b,i} y_{i,b,c}} \tag{1}$$

We modify the loss function to be calculated only for classes that are annotated in the corresponding partially labeled dataset [21,4]. This partially loss formalize as follow:

$$L = \frac{1}{\sum_{b,c} 1_{b,c}^{(h)}} \sum_{b,c} \left( \frac{1_{b,c}^{(h)}}{I} \sum_i BCE(\hat{y}_{i,b,c},\ y_{i,b,c}) - \frac{2 \sum_i 1_{b,c}^{(h)} \hat{y}_{i,b,c}\, y_{i,b,c}}{\sum_i 1_{b,c}^{(h)} \hat{y}_{i,b,c} + \sum_i 1_{b,c}^{(h)} y_{i,b,c}} \right) \tag{2}$$

$$1_{b,c}^{(h)} = \begin{cases} 0,\ if\, c \in S\, and\, h = False, \\ 1,\ otherwise, \end{cases}$$

where $c \in S$ is the selected class set, we set $S = \{tumor\}$, $h$ is false if the ground truth data does not include the class $c$, otherwise it is true. The loss use the summation between dice loss and binary cross entropy loss because compound loss functions have been proved to be robust in various medical image segmentation tasks [15].

**Table 1.** Network architecture and inference process.

| | |
|---|---|
| Channels in the first stage | 16 |
| Convolution number per stage | 2 |
| Patch size | 128×128×128 |
| Downsampling times | 4 |
| inference process | (Sigmoid, Threshold, Resample) |
| Deep supervision | True |

**Speeding inference** In order to improve inference speed and reduce resource consumption, we use a small-size network structure in reference [10]. And we change the default resampling function and order, which effectively speeds up the inference. The setup of network architecture and inference process are presented in Table 1. Comparison of different strategy settings in Table 2 . The default is full resolution setting of nnU-Net and the small is low resolution modified. The tiny is the first stage of the cascade network that we design to have a lower resolution. However, we do not use the cascade network as the final docker submission because it does not improve the accuracy and speed of the segmentation results.

**Table 2.** Comparison of different strategy settings. The order of axes of input patch size and spacing is (z,y,x).

| Settings | Default | Small | Tiny |
|---|---|---|---|
| Channels in the first stage | 32 | 16 | 8 |
| Convolution number per stage | 2 | 2 | 2 |
| Patch size | 56×192×160 | 128×128×128 | 80×96×96 |
| Downsampling times | 5 | 4 | 4 |
| Input spacing | (2.5, 0.8, 0.8) | (2.42, 1.95, 1.95) | (5, 3.9, 3.9) |

### 2.3   Post-processing

We do not perform any post-processing, such as connected component analysis or testing time augmentation, during our inference.

## 3   Experiments

### 3.1   Dataset and evaluation measures

The FLARE 2023 challenge is an extension of the FLARE 2021-2022 [17][18], aiming to aim to promote the development of foundation models in abdominal disease analysis. The segmentation targets cover 13 organs and various abdominal lesions. The training dataset is curated from more than 30 medical centers under the license permission, including TCIA [2], LiTS [1], MSD [23], KiTS [8,9], autoPET [7,6], TotalSegmentator [28], and AbdomenCT-1K [19]. The training set includes 4000 abdomen CT scans where 2200 CT scans with partial labels and 1800 CT scans without labels. The validation and testing sets include 100 and 400 CT scans, respectively, which cover various abdominal cancer types, such as liver cancer, kidney cancer, pancreas cancer, colon cancer, gastric cancer, and so on. The organ annotation process used ITK-SNAP [29], nnU-Net [11], and MedSAM [16].

The evaluation metrics encompass two accuracy measures—Dice Similarity Coefficient (DSC) and Normalized Surface Dice (NSD)—alongside two efficiency measures—running time and area under the GPU memory-time curve. These metrics collectively contribute to the ranking computation. Furthermore, the running time and GPU memory consumption are considered within tolerances of 15 seconds and 4 GB, respectively.

### 3.2   Implementation details

**Environment settings** The development environments and requirements are presented in Table 3.

Table 3. Development environments and requirements.

| System | Ubuntu 20.04.5 LTS |
|---|---|
| CPU | Intel(R) Xeon(R) Gold 6354 CPU @ 3.00GHz |
| RAM | 16×4GB; 1600MT/s |
| GPU (number and type) | 1 × NVIDIA A100 40G |
| CUDA version | 11.7 |
| Programming language | Python 3.10.11 |
| Deep learning framework | Pytorch 2.0.0, torchvision 0.2.2 |
| Specific dependencies | nnU-Net 2.0 |
| Code | https://github.com/orangeqqq/FLARE23 |

**Training protocols** The training protocols of the small nnU-Net are listed in Table 4. For the unlabeled images, we select 100 cases with the pseudo label to train the network. For partial labels, we use the partial cross-entropy and dice

**Table 4.** Training protocols.

| | |
|---|---|
| Network initialization | "He" normal initialization |
| Batch size | 4 |
| Patch size | 128×128×128 |
| Total epochs | 1000 |
| Optimizer | SGD with nesterov momentum ($\mu$ =0.99) |
| Initial learning rate (lr) | 0.01 |
| Lr decay schedule | Poly learning rate policy: $(1 - epoch/1000)^{0.9}$ |
| Training time | 10 hours |
| Loss function | Cross entropy loss and dice loss |
| Number of model parameters | 5.22M[1] |
| Number of flops | 121G[2] |
| $CO_2$eq | 11.2 Kg[3] |

loss in the training stage. the pseudo labels generated by the FLARE22 winning algorithm [10] and the best-accuracy-algorithm [27]. We employ the same data augmentation as the default setting of nnU-Net, which includes additive brightness, gamma, rotation, scaling, and elastic deformation on the fly during training. During inference, the model does not perform test time augmentation (TTA) of flipping. The patch sampling strategy is foreground over-sampling. Finally, we choose the model that obtains the fast and best accuracy on the online validation.

## 4   Results and discussion

**Table 5.** Quantitative evaluation results in terms of DSC(%) and NSD(%).

| Target | Public Validation | | Online Validation | | Testing | |
|---|---|---|---|---|---|---|
| | DSC | NSD | DSC | NSD | DSC | NSD |
| Liver | 95.54 ± 2.53 | 96.86 ± 5.34 | 95.62 | 97.12 | 94 | 95.75 |
| Right Kidney | 87.89 ± 19.54 | 88.35 ± 20.41 | 89.35 | 90.01 | 90.64 | 90.44 |
| Spleen | 93.06 ± 3.77 | 93.55 ± 8.12 | 93.18 | 93.86 | 95.23 | 93.1 |
| Pancreas | 82.05 ± 5.93 | 95.41 ± 4.86 | 80.72 | 94.5 | 82.34 | 95.29 |
| Aorta | 93.05 ± 2.06 | 97.64 ± 3.19 | 93.35 | 97.98 | 93.03 | 98.25 |
| Inferior vena cava | 88.05 ± 5.56 | 90.98 ± 6.52 | 88.06 | 90.7 | 88.84 | 92.31 |
| Right adrenal gland | 74.67 ± 12.86 | 91.33 ± 13.74 | 75.24 | 92.12 | 72.30 | 90.79 |
| Left adrenal gland | 71.41 ± 13.29 | 88.43 ± 14.0 | 72.83 | 89.22 | 70.88 | 88.32 |
| Gallbladder | 82.06 ± 19.92 | 81.27 ± 21.06 | 82.52 | 81.86 | 74.83 | 74.85 |
| Esophagus | 78.46 ± 14.01 | 91.15 ± 14.41 | 79.12 | 92.15 | 81.91 | 94.91 |
| Stomach | 90.23 ± 6.08 | 95.25 ± 6.71 | 90.6 | 95.07 | 89.59 | 94.53 |
| Duodenum | 78.06 ± 8.28 | 93.96 ± 5.57 | 78.25 | 93.53 | 79.30 | 94.42 |
| Left kidney | 86.96 ± 16.61 | 87.77 ± 17.72 | 87.96 | 88.78 | 89.12 | 89.19 |
| Tumor | 18.21 ± 23.28 | 10.27 ± 15.24 | 15.14 | 8.72 | 17.61 | 8.32 |
| Average | 79.98 ± 10.98 | 85.87 ± 11.21 | 80.14 | 86.12 | 79.97 | 85.75 |

### 4.1   Quantitative results on validation set

In Table 5, we report the DSC and NSD of the final docker commit results. The average of the 50 public validation and the 100 online validation are the same, both achieving a DSC of about 0.80 and an NSD of 0.86. In general, large organs like the liver, spleen, kidney, and stomach have high accuracy. However, accurate identification of small and complex objects, such as tumors, adrenal glands, and the duodenum, presents significant challenges. It requires more attention, especially when dealing with extremely small and indistinct boundaries.

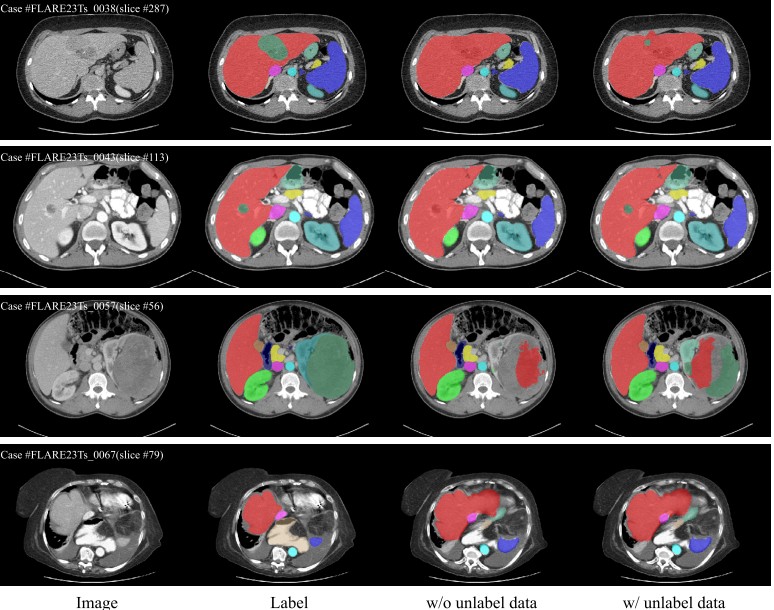

**Fig. 2.** Qualitative results on two easy cases (Case #FLARE23Ts_0038 with DSC of 0.89 and Case #FLARE23Ts_0043 with DSC of 0.84) and two hard cases (Case #FLARE23Ts_0057 with DSC of 0.66 and Case #FLARE23Ts_0067 with DSC of 0.74).

We report the online validation results of the model without unlabelled data, normal inference processes, and cascade networks in Table 7. The model using unlabelled data resulted in an increase of the DSC from 0.7925 to 0.8013. Specifically, in tumor regions, it increased the DSC by 0.045. Additionally, normal inference alone increased the DSC by approximately 0.04. However, the cascade network, P-Cascade and N-Cascade, which added a network training in a lower resolution setup with twice the spacing of the original, did not achieve higher DSC and NSD results. P-Cascade is the results of partially compound loss and N-Cascade is the the results of normal compound loss. Comparing the

two, we find that the model trained by partially labeled loss has better results for tumor segmentation, with an improvement in DSC value of 0.05.

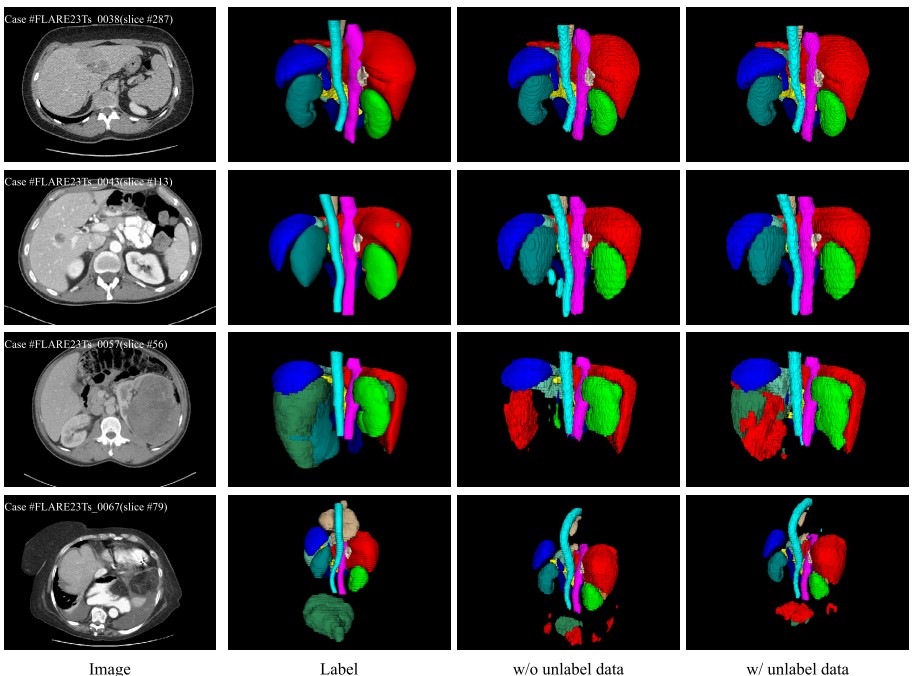

**Fig. 3.** 3D visualization on two easy cases (Case #FLARE23Ts_0038 with DSC of 0.89 and Case #FLARE23Ts_0043 with DSC of 0.84) and two hard cases (Case #FLARE23Ts_0057 with DSC of 0.66 and Case #FLARE23Ts_0067 with DSC of 0.74).

### 4.2    Qualitative results on validation set

Figure 2 presents easy and difficult validation set examples for segmentation, along with a 3D visualization in Figure 3. Promising results were observed for Case #FLARE23Ts_0038 and Case #FLARE23Ts_43, but the segmentation of Case #FLARE23Ts_57 and Case #FLARE23Ts_67 was poor due to a large tumor that caused the network to make classification errors.

### 4.3    Segmentation efficiency results on validation set

In Table 6, we observe a set of cases that increase in size from (512,512,55) to (512,512,554). The efficiency evaluation results are reported from official tests. It is seen that the average max GPU is 2531MB, and run time increase twice

for the biggest case #0029 than the smallest case #0001. This demonstrates the effectiveness of our inference strategy.

**Table 6.** Quantitative evaluation of segmentation efficiency in terms of the running time and GPU memory consumption. Total GPU denotes the area under GPU Memory-Time curve. Evaluation GPU platform: NVIDIA QUADRO RTX5000 (16G).

| Case ID | Image Size | Running Time (s) | Max GPU (MB) | Total GPU (MB) |
|---|---|---|---|---|
| 0001 | (512, 512, 55) | 19.61 | 2426 | 10028 |
| 0051 | (512, 512, 100) | 17.83 | 2590 | 12296 |
| 0017 | (512, 512, 150) | 30.86 | 2634 | 15949 |
| 0019 | (512, 512, 215) | 22.72 | 2486 | 12401 |
| 0099 | (512, 512, 334) | 27.94 | 2586 | 15394 |
| 0063 | (512, 512, 448) | 33.50 | 2630 | 17508 |
| 0048 | (512, 512, 499) | 35.22 | 2614 | 18610 |
| 0029 | (512, 512, 554) | 42.53 | 2744 | 22299 |

### 4.4   Results on final testing set

In table 5, we report the DSC and NSD of the final testing set. The average values are comparable to those of the 50 public validations and the 100 online validations, with both achieving a DSC of about 0.80 and a NSD of about 0.86. In general, the low accuracy of segmenting small and complex shaped objects such as tumors, adrenal glands and duodenums Their accurate segmentation still faces great challenges and needs more attention, especially when dealing with extremely small and unclear boundaries.

### 4.5   Limitation and future work

There are many ways to improve the network inference process, such as a more efficient sliding window. The challenge provided 4,000 CT cases, but we only utilized 300 cases and did not adequately utilize the data. For the challenging task of tumor segmentation, pseudo-labeling is a simple and effective way to improve model performance, and we will continue to explore methods that utilize both pseudo-labeling and partial labeling learning in the future.

## 5   Conclusion

In this paper, we present a framework that combines partial labeling learning and pseudo-labeling, which is effective and flexible for a variety of situations. In addition, we use a small nnU-Net and improve the inference process, effectively reducing its required computational resources and inference time. Because the amount of data used in training is small, performance on the full data will be explored in the future. The approach in this paper will be a good baseline result for exploring partial labeling learning and pseudo-labeling.

**Table 7.** Ablation studies of online validation quantitative evaluation results in terms of DSC(%) and NSD(%). P-Cascade is the results of partially compound loss and N-Cascade is the the results of normal compound loss.

| Target | w/o unlabeled data | | Normal inference | | N-Cascade | | P-Cascade | |
|---|---|---|---|---|---|---|---|---|
| | DSC | NSD | DSC | NSD | DSC | NSD | DSC | NSD |
| Liver | 95.77 | 97.09 | 97.34 | 97.46 | 95.63 | 97.63 | 95.9 | 97.5 |
| Right Kidney | 89.93 | 90.49 | 92.18 | 91.46 | 90.27 | 91.27 | 89.9 | 91.28 |
| Spleen | 93.57 | 94.46 | 97 | 97.58 | 91.34 | 92.01 | 92.68 | 93.44 |
| Pancreas | 79.66 | 93.5 | 84.22 | 94.82 | 79.74 | 93.78 | 79.79 | 93.6 |
| Aorta | 92.29 | 96.86 | 96.57 | 99.03 | 92.59 | 97.42 | 93.23 | 97.79 |
| Inferior vena cava | 87.24 | 89.84 | 91.06 | 91.43 | 86.38 | 88.25 | 87.06 | 89.12 |
| Right adrenal gland | 74.24 | 91.78 | 85.51 | 95.48 | 72.75 | 90.19 | 73.35 | 90.59 |
| Left adrenal gland | 71.19 | 87.59 | 83.27 | 93.27 | 72.47 | 89.09 | 72.33 | 88.76 |
| Gallbladder | 80.34 | 79.38 | 86.09 | 86.55 | 77.9 | 77.05 | 80.54 | 79.83 |
| Esophagus | 78.13 | 90.88 | 83.09 | 93.4 | 78.57 | 91.89 | 79.05 | 92.26 |
| Stomach | 90.52 | 94.52 | 93.12 | 95.51 | 89.95 | 94.58 | 90.37 | 94.91 |
| Duodenum | 77.31 | 93.19 | 81.45 | 93.43 | 78.42 | 94.19 | 78.25 | 93.97 |
| Left kidney | 88.69 | 88.97 | 91.06 | 90.65 | 88.23 | 89.43 | 87.67 | 86.93 |
| Tumor | 10.64 | 5.92 | 15.17 | 8.42 | 10.25 | 6.99 | 15.88 | 10.43 |
| Average | 79.25 | 85.32 | 84.08 | 87.75 | 78.89 | 85.27 | 79.71 | 85.74 |

**Acknowledgements** The authors of this paper declare that the segmentation method they implemented for participation in the FLARE 2023 challenge has not used any pre-trained models nor additional datasets other than those provided by the organizers. The proposed solution is fully automatic without any manual intervention. We thank all the data owners for making the CT scans publicly available and CodaLab [20] for hosting the challenge platform.

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

**Table 8.** Checklist Table. Please fill out this checklist table in the answer column.

| Requirements | Answer |
|---|---|
| A meaningful title | Yes |
| The number of authors (≤6) | 4 |
| Author affiliations and ORCID | Yes |
| Corresponding author email is presented | Yes |
| Validation scores are presented in the abstract | Yes |
| Introduction includes at least three parts: background, related work, and motivation | Yes |
| A pipeline/network figure is provided | 1 |
| Pre-processing | 3 2.1 |
| Strategies to use the partial label | 3 2.2 |
| Strategies to use the unlabeled images. | 3 2.2 |
| Strategies to improve model inference | 4 2.2 |
| Post-processing | 4 2.3 |
| Dataset and evaluation metric section is presented | 5 3.1 |
| Environment setting table is provided | 3 |
| Training protocol table is provided | 4 |
| Ablation study | 7 |
| Efficiency evaluation results are provided | 6 |
| Visualized segmentation example is provided | 2 |
| Limitation and future work are presented | Yes |
| Reference format is consistent. | Yes |