# OpenReview forum: "Selected Partially Labeled Learning for Abdominal Organ and Pan-cancer Segmentation"
_MICCAI.org/2023/FLARE — Submitted to FLARE 2023_

### Official Review · Reviewer_qaqC · 2023-09-26
**review for 'Selected Partially Labeled Learning for Abdominal Organ and Pan-cancer Segmentation'**

**Rating:** 5
**Confidence:** 5

**Review:**

Pros: The paper is in a good layout, according to the official guidelines. The author ingeniously applied the pseudo-labels generated by two methods and used partial loss.

Cons: There are still some problems.
1.	The overview of the framework is still not that clear.
2.	The corresponding author should be indicated.
3.	Please check the manuscript carefully. For example, there are some typos in the "Method" section. 'the' should be capitalized as 'The'. ‘verifications’ should be ‘Validation’.
4.	Consistency in the tense used throughout the manuscript. For instance, use the present simple tense in section 2.2.
5.	The case number in Section 4.2 and the caption of Figures 2 and 3 should be the same with the number in Figures 2 and 3.
6.	The performance is not promising, maybe because of the small network and small patch size setting.

---

### Official Review · Reviewer_X7kx · 2023-09-29
**Review for "Selected Partially Labeled Learning for Abdominal Organ and Pan-cancer Segmentation."**

**Rating:** 5
**Confidence:** 5

**Review:**

The paper is well-organized and presents pseudo-label generation and a partial loss. And its method achieves a good baseline performance.

However, there are some concerned issues in this paper.
Only 300 cases are utilized in the study. How do you select these cases included in the study? just randomly? When producing pseudo-label, which result produced by two FLARE 2022 methods do you utilize? These issues should be elucidated.
In my opinion, the training of nnU-net in the study is a fully supervised manner. Author didn't train it iteratively? Furthermore, the effects of the original loss and modified partial loss should be compared.

Other issue,
Only one author is listed, and the corresponding author should be indicated.

---

### Official Review · Reviewer_zUc9 · 2023-10-16
**Review for "Selected Partially Labeled Learning for Abdominal Organ and Pan-cancer Segmentation."**

**Rating:** 5
**Confidence:** 5

**Review:**

The paper proposes compound loss for training neural networks on partially labeled medical images. Achieves segmentation with a Dice Similarity Coefficient of 0.8514 and  0.1514. Improves inference speed, using 2531 MB of maximum GPU memory. However, there are some issues that need to be addressed:
1. This paper only uses 300 cases. The selection criteria and rationale should be clarified, and the impact of data size on experimental performance should be analyzed.
2. One focus of the work is the lightweight adaptation of nnU-Net. A comparison between the original and modified configurations would help readers understand the improvements made.
3. Similar to point 2, comparing the inference time before and after optimization would provide clarity on the extent of the improvements.
4. Some English expressions in the paper may need improvement.

---

### Decision · Program_Chairs · 2023-10-24

Accept